# Psychotherapy with Suicidal Patients: The Integrative Psychodynamic Approach of the Boston Suicide Study Group

**DOI:** 10.3390/medicina55060303

**Published:** 2019-06-24

**Authors:** Mark Schechter, Elsa Ronningstam, Benjamin Herbstman, Mark J. Goldblatt

**Affiliations:** 1North Shore Medical Center, 81 Highland Avenue, Salem, MA 01970, USA; 2Harvard Medical School, 25 Shattuck Street, Boston, MA 02115, USA; ronningstam@email.com (E.R.); bherbstman@mclean.harvard.edu (B.H.); mark_goldblatt@hms.harvard.edu (M.J.G.); 3McLean Hospital, 115 Mill Street, Belmont, MA 02478, USA

**Keywords:** psychodynamic psychotherapy, integrative, suicide, therapeutic alliance, countertransference, hope

## Abstract

Psychotherapy with suicidal patients is inherently challenging. Psychodynamic psychotherapy focuses attention on the patient’s internal experience through the creation of a therapeutic space for an open-ended exploration of thoughts, fears, and fantasies as they emerge through interactive dialogue with an empathic therapist. The Boston Suicide Study Group (M.S., M.J.G., E.R., B.H.), has developed an integrative psychodynamic approach to psychotherapy with suicidal patients based on the authors’ extensive clinical work with suicidal patients (over 100 years combined). It is fundamentally psychodynamic in nature, with an emphasis on the therapeutic alliance, unconscious and implicit relational processes, and the power of the therapeutic relationship to facilitate change in a long-term exploratory treatment. It is also integrative, however, drawing extensively on ideas and techniques described in Dialectical Behavioral Therapy (DBT), Mentalization Based Treatment (MBT), Cognitive-Behavioral Therapy (CBT), as well on developmental and social psychology research. This is not meant to be a comprehensive review of psychodynamic treatment of suicidal patients, but rather a description of an integrative approach that synthesizes clinical experience and relevant theoretical contributions from the literature that support the authors’ reasoning. There are ten key aspects of this integrative psychodynamic treatment: 1. Approach to the patient in crisis; 2, instilling hope; 3. a focus on the patient’s internal affective experience; 4. attention to conscious and unconscious beliefs and fantasies; 5. improving affect tolerance; 6. development of narrative identity and modification of "relational scripts"; 7. facilitation of the emergence of the patient’s genuine capacities; 8. improving a sense of continuity and coherence; 9 attention to the therapeutic alliance; 10. attention to countertransference. The elements of treatment are overlapping and not meant to be sequential, but each is discussed separately as an essential aspect of the psychotherapeutic work. This integrative psychodynamic approach is a useful method for suicide prevention as it helps to instill hope, provides relational contact and engages the suicidal patient in a process that leads to positive internal change. The benefits of the psychotherapy go beyond crisis intervention, and include the potential for improved affect tolerance, more fulfilling relational experiences, emergence of previously warded off experience of genuine capacities, and a positive change in narrative identity.

## 1. Introduction

Psychodynamic psychotherapy focuses attention on the patient’s internal experience through the creation of a therapeutic space for an open-ended exploration of thoughts, fears, and fantasies as they emerge through interactive dialogue with an empathic therapist. The Boston Suicide Study Group (M.S., M.J.G., E.R., B.H.) has developed an integrative psychodynamic approach to psychotherapy with suicidal patients based on the authors’ extensive clinical work with suicidal patients. It is fundamentally psychodynamic in nature, with an emphasis on the therapeutic alliance, unconscious and implicit relational processes, and the power of the therapeutic relationship to facilitate change in a long-term exploratory treatment. However, this is also an integrative approach, drawing extensively on DBT, MBT, CBT, and developmental and social psychology research. The authors have identified ten key aspects of their treatment approach: approach to the patient in crisis; instilling hope; a focus on the patient’s internal affective experience; attention to conscious and unconscious beliefs and fantasies; improving affect tolerance; development of narrative identity and modification of "relational scripts"; facilitation of the emergence of the patient’s genuine capacities that have been thwarted in the course of development; improving a sense of continuity and coherence; attention to the therapeutic alliance; and attention to countertransference. These elements of treatment are overlapping and are not meant to be sequential. In fact, the therapist and patient may at times be engaged in working in multiple areas fluidly and simultaneously in the course of treatment. This paper discusses each of these areas, making use of relevant literature and clinical examples to support the authors’ reasoning.

Psychotherapy with suicidal patients is inherently extremely challenging. First, the therapist must accept that despite his or her best efforts there is no way to definitively control the outcome, and a patient may complete suicide. Suicidal patients are suffering terribly, and therapists are generally highly motivated to help but often lack any tools to alleviate the suffering (at least in the immediate term). Hopelessness can be particularly difficult to bear, and the therapist may feel worn down over time, beginning to accede to the patient’s often stated belief that nothing can possibly be of help. In order to instill hope the therapist must have his or her own roadmap for how psychotherapy can help the patient to improve. It is our hope that this paper will help psychotherapists to understand our integrative psychodynamic approach and the ways in which it can help the patient move from suicidal despair to a life that feels worth living.

## 2. Background

The authors of this paper represent the Boston Suicide Study Group. We are all psychoanalysts with extensive experience in clinical work with suicidal patients (over 100 years combined) and an interest in what can be learned from integrating ideas and techniques from other psychotherapeutic approaches. We have studied and treated many psychiatric patients who have presented with suicidal ideation and plans, among them many who have made serious suicide attempts as well as some completed suicides. We have also studied the literature on psychotherapies for suicidal patients and on psychodynamic psychotherapy. This paper has been written with the intent to synthesize what we have found to be the key elements and successful treatment with suicidal patients, and to support our reasoning with relevant literature and clinical examples. It is not meant to be a comprehensive, systematic review of the literature on psychodynamic psychotherapy with suicidal patients, but rather an opportunity for us to share our integrative psychodynamic approach.

## 3. Elements of the Integrative Psychodynamic Approach

### 3.1. Approach to the Patient in Crisis

A thorough safety evaluation is essential when a patient is in an acute suicidal crisis. The therapist needs to assess the degree of imminent risk and to determine whether measures beyond psychotherapy are required to keep the patient safe (e.g., emergency department, psychiatric hospitalization, etc.). Reviews of evidence-based treatments of chronic suicidality have found that all recommend a clear frame for the treatment and a defined strategy for managing suicidal crises (i.e., a crisis plan) [1,2]. Early establishment of a treatment alliance is facilitated by the therapist’s attitude of non-judgmental acceptance and validation of the patient’s experience [3,4,5].

Patients generally enter to the clinical encounter feeling terrible about themselves, with feelings of shame, harsh self-criticism/self-attack, and a sense of hopelessness and/or failure. The therapist’s empathic ability to communicate an understanding of the ways in which the patient’s crisis and suicidality are understandable, given his or her situation, past experiences and current internal state, forms the foundation of the alliance. Validation helps to mitigate the experience of loneliness and aloneness, decreases self-blame and harsh self-attack, and models a hopeful attitude about the possibility that the patient can be understood and ultimately helped.

The therapist is interested in the patient’s "reasons for living" [6,7,8], including the patient’s important relationships such as family (in particularly children), to get a sense of what the patient feels there is in life to live *for*. The therapist is in a unique position to engage into a respectful dialogue with the patient, not necessarily accepting the patient’s automatic defenses as the final answer. It is not uncommon, for example, for patients to say that they believe that their children "will be better off without me”, as well as describe themselves as "a burden" to loved ones. With gentle confrontation and exploration, one finds that some patients actually hold more conflicted feelings and beliefs but are locked in on these explanations because it protects them from bearing the pain of their conflict and from the genuine consequences that their suicide would engender. An iterative dialogue with the therapist, in which the therapist explores and questions the patient’s expressed beliefs, opens up the possibility of modification. The therapist’s interest in what the patient thinks and feels about suicide and death can also be experienced as a validation, that the patient is worth the therapist’s interest and engagement. The goal is not to try to control the patient’s beliefs, but rather to enable the patient to experience suicidal behavior as less desirable, acceptable, and effective than the alternative of working on coping and managing distress. This can help to tip the balance away from self-destructive action [8,9].

### 3.2. Instilling Hope

Patients who struggle with despair and suicidality often come to the therapeutic encounter with feelings of shame, discouragement, and hopelessness. They have no "road map," of how they could possibly get from where they are feeling now to building a life worth living. The therapist can help to provide such a “road map”, based on an understanding of the gains that can be achieved in the psychotherapeutic process. The patient sustains hope through the therapeutic relationship, symptom management and validation of a core identity [10], and this facilitates a gradual change in perspective on life and living.

Snyder [11] described hope as requiring both a sense of agency (sense that one has some capacity to affect positive change) and pathways (a sense of some possible routes for moving forward). The therapist holds both of these components of hope and communicates them both implicitly and directly to the patient. This helps the patient to begin to experience a sense of agency, and to have some glimmer of how he or she can survive the current feelings and go on to have a life worth living. Helping to sustain the therapist’s hope is the data that suicidal behavior is often a transient phenomenon. Studies have found that the vast majority of people who survive even the most severe suicide attempts actually do not go on to kill themselves [12,13]. Psychotherapy has been found to reduce the risk of recurrent suicide attempts and self-injury [14].

Psychotherapy can help support the patient who is engaging in an internal process that can gradually lead to a changed perspective and greater hope for the future. One aspect of this process is a genuine acceptance of the actual conditions that precipitated the suicidal crisis. This is especially important for patients who have faced drastic changes in life (e.g., losses, humiliations, etc.), or are coming to a realization that life is not going to turn out as expected or envisioned. Related to this is a person’s ability to tolerate and accept loss, sadness, and disappointment. This involves a process of grief and mourning for what one had and will never have again, and/or what could have been but will not be. It requires engagement of the patient’s self-reflective capacity, which is facilitated in psychotherapy by validation of the patient’s pain, modeling reflection, affective engagement and modulation of the patient’s harsh self-criticism/self-attack and feelings of failure.

Over time, the therapist can support the patient in moving from a fixation on what is ideal, to recognizing the possibility of what can be good enough. Early in the process, the "good enough" is often devalued and depicted as not worth living for, but with ongoing reflection, mourning, and support, the patient can learn that it is possible to find meaning and even joy in life with goals and expectations that have shifted in line with what is possible. This includes a re-engagement in relationships and a greater sense of belonging. At the same time, mobilization of the patient’s sense of agency and instilling of hope for the future can lead to actions on the patient’s part that can have a positive impact on relationships and circumstances. This can create a positive feedback loop that can lead to actual achievements beyond what the patient could have envisioned in a prior state of hopelessness.

### 3.3. Focus on Internal Experience and Affect

A distinguishing aspect of psychodynamic psychotherapy is the close attention to the patient’s internal experience. Shneidman eloquently described suicide as ‘a combined movement toward cessation and away from intolerable, unendurable, unacceptable anguish’ ([15], p. 6). Multiple other terms have been used in the literature to describe this affective state by many differing theoretical orientations, trying to capture essentially the same experience: It has been variously referred to as “desperation” [16], “mental pain” [17], “psychache” [18], “emotional dysregulation” [3], and “annihilation anxiety” [19]. Hendin et al. [16] found that therapists reported a higher number of intense, agonizing affective states in patients who completed suicide in the course of psychotherapy, as compared with a severely depressed, non-suicidal comparison group. The most frequently cited affect was desperation, defined as a state of anguish accompanied by an urgent need for relief. More recently, Galynker [20] has focused on “entrapment”, an emotional experience of desperation with no perceived way out.

The state of "aloneness" has been described as an unbearable experience that makes people feel desperate and puts them at risk for suicidal behavior [21,22]. Joiner and colleagues have studied a closely related experience, that of "thwarted belongingness", and found it to be a key factor in making completion of suicide possible [23,24]. Maltsberger [22] described aloneness as “an experience beyond hope…. This anxiety is the anxiety of annihilation—panic and terror. People will do anything to escape from this experience” (p. 50). The experience can feel timeless, as though once it has started it will continue for every, which adds to the sense of desperation. Aloneness is qualitatively different from loneliness, which is contingent and time limited. It is the loss of capacity to *experience* the closeness and caring of others, even if they are present and available.

In the face of an unrelenting, unbearable affective experience, cognition changes in a way that makes rational problem solving much more difficult. Baumeister [25] described the suicidal patient as moving into a state of cognitive “deconstruction” (pp. 92–93), characterized by rigidity, increasingly concrete thinking. a narrowing of sense of time to the present and an exclusive focus on immediate goals, and a lack of integration and meaning. Linehan [3] has observed that in states of severe emotional dysregulation, patients often lose certain capacities that they have in a calmer state of mind. Thinking becomes much more concrete and constricted; when the patient sees no alternative, no endpoint or escape, the risk of suicidal behavior is increased. Maltsberger [26] described the traumatic state of disintegration that accompanies a suicidal state.

An unbearable affective experience is distinct from depression. For vulnerable patients, it can come on suddenly, in response to a stressor, at times without the patient making the connection between stress and response. It can also be relieved transiently if the patient is suddenly provided a respite, such as by psychiatric hospitalization. The relief, however, can be suddenly and overwhelmingly reversed when the patient is re-exposed to overwhelming stressors at discharge, which is probably a major factor in the high rate of post-discharge suicide [27]. Ideally, psychotherapeutic efforts in the hospital can be targeted toward helping the patient to anticipate and problem-solve for upcoming crises, and to develop a usable and well-practiced crisis plan. It is important for the therapist to be aware that the immediate period post-discharge is very high risk for the patient, and to intervene accordingly.

### 3.4. Attention to Patients Conscious and Unconscious Beliefs and Fantasies

Psychodynamic psychotherapy provides an opportunity to explore the patient’s conscious and unconscious beliefs and fantasies about suicide. These fantasies are often not rational and in contradiction with the patient’s known reality. The idea of suicide can have very different meanings and can be facilitated or mitigated by different sets of fantasies. Some may be conscious, and others may be out of their awareness. These beliefs and fantasies can be wide ranging, and at times in contradiction with each other and/or with reality. Some patients believe they deserve punishment; others long for a fantasized reunion with lost parents or a loved one; some have a fantasy that they will get to “see” the reaction of others to the suicide, and even that they will experience pleasure in doing so, even while knowing rationally that this is not the case; some experience their body as hateful and want to destroy it, perhaps because it is identified with a past abuser (see [28]). 

The patient’s fantasies about suicide and death can influence the likelihood of suicidal action. Articulating and exploring these fantasies enables a therapeutic dialogue which gives the therapist the opportunity to respectfully challenge certain automatic assumptions. For example, perhaps the mother who states the belief that her child would be better off without her, is in fact able and willing to consider the possibility that she might be misreading the situation because of her depression and despair. A man who is experiencing aloneness and has a fantasy of rejoining his deceased wife in heaven may be willing to consider the possibility that suicide might *not* mean a blissful reunion, and that there are other consequences that he is not letting himself think through. The therapist’s interest in exploring the patient’s fantasies demonstrates his or her wish to get a full understanding of the patient’s experience, and in this way may bolster the patient’s sense that he or she matters. It also provides an opportunity for gradual modification of these beliefs in the context of an iterative dialogue over time.

Some suicidal contemplation and fantasy can be “self-sustaining”. Rather than facilitating action, these thoughts can help to calm people down and help them regulate affect in a way that helps preclude the need to act on suicide [29]. These are often patients for whom the idea that they could complete suicide has served as a psychological "out" which gives them a feeling of *potential* escape from their distress; as long as they hold onto suicide as an option, they do not need to act. This kind of suicidal contemplation is “a self-soothing measure and an aid to narcissistic cohesion” (p. 612). Patients often do not understand this distinction and may be very troubled by their suicidal thoughts even as it serves an important function for them. The therapist can reassure the patient about the role of suicidal fantasy, and that thoughts do not equate with action. In fact, in the general population the ratio of those who harbor suicidal thoughts to those who kill themselves is over 200:1 [30].

### 3.5. Improving Affect Tolerance

Increasing affect tolerance is central to psychotherapy with patients with emotional dysregulation. Affect tolerance might roughly be defined as increasing one’s capacity to think in the presence of strong emotions, and to bear feelings without having to suppress, dissociate, or act impulsively. In Dialectical Behavior Therapy (DBT), emotion regulation and distress tolerance skills are targeted to increase affect tolerance [3]. The therapist may integrate teaching the patient these skills in a psychodynamic treatment frame.

Perhaps unique to psychodynamic therapy is the opportunity to work with the patient on unconscious fears and avoidances that have a profound effect on the patient’s sense of self, transferences, core beliefs, and relationships. The patient brings his or her sense of shame, humiliation, and feelings of despair to the treatment, often unconsciously fearing negative reaction on the part of the therapist that comes from a template of past relationships. Ideally with validation and support, these feelings can gradually extinguish, giving the patient more flexibility and choice. People are unaware of automatic avoidance defenses and are highly motivated to continue to use them to ward off anxiety provoking and conflictual thoughts and feelings. This keeps such material out of consciousness, and unavailable to be worked on in therapy. When the therapist gently notices avoidant defenses (e.g., "when I said that, I noticed you averted your eyes and started to change the subject…what was going on there?"), it offers the patient an opportunity to consider what he or she is warding off. At times, the therapist may use explicit exposure techniques (e.g., “can you try saying that again while looking me in the eye, and just sit with what you are feeling?”) in order to facilitate the patient’s exposure to underlying uncomfortable feelings that are being warded off. With ongoing therapy, the need for these automatic avoidant defenses gradually begins to lessen. The extinguishing of shame, fear, expectancies of harsh attacks/blame, etc., allows for greater tolerance of affect, increasing the patient’s flexibility and freedom of expression. This work is closely related to work on narrative identity (described below), with the working through of the patient’s “relational scripts” that have led to recurrent experiences that sustain negative ideas/beliefs/conclusions about self.

### 3.6. Narrative Identity and "Relational Scripts”

The development of personal “narratives” has been found to be critical to the establishment and ongoing development of identity [31,32,33,34,35,36,37] People tell themselves stories to link experiences, make meaning from events, establish a sense of continuity of experience, and articulate ways in which they have changed [38]. In this way, people develop a “life story,” described as “an internalized and evolving narrative of the self that selectively reconstructs the past and anticipates the future in such a way as to provide a life with an overall sense of coherence and purpose” ([36], p. 1372). The personal narratives of the depressed or suicidal patient’s self-narrative are generally harshly self-critical and negative (e.g., “Everything I try to do fails I never get things right”). Galynker described the “suicidal narrative” as telling the story of “a present that is so intolerable that the future becomes unimaginable” ([20], p. 31).

Personal narratives are actually co-constructed, with the listener influencing whether and how the story is told, and meanings that can be derived from it [33,39,40]. Sharing one’s personal stories with others helps to make meaning from experiences, and to try out ways of understanding oneself. Psychodynamic psychotherapy provides an opportunity for the patient to engage with a new “listener,” an active participant who engages in the development of the patient’s narrative [37]. The therapist notices and takes up the harsh self-criticism and self-attack embedded in the patient’s story, and in so doing begins a process of modification.

Patients often begin treatment with a narrative that does not “connect the dots” between past experience and current despair. A patient might tell a story of severe neglect and/or abuse that clearly would lead to problems with self-esteem and trust but not make this connection. Together, therapist and patient engage in an iterative dialogue in which the patient’s narrative is reworked and co-constructed. Not all patients come into treatment with the capacity to share a coherent narrative, particularly in the context of neglectful and abusive early experiences. Problems may occur at the “pre-narrative” and “proto-narrative” levels at which mental images are generated and linked to emotions, which eventually become the building blocks for higher level, verbal, narrative construction [41]. Early trauma and neglect can lead to a lack of emotional marking of mental images and an inability to recruit these images into meaningful sequences that integrate cognition and emotion. As a result, narratives are disorganized, impoverished, and sometimes frankly incoherent, and “the therapist finds it difficult to comprehend the patient’s mental state” (p. 236). This leads to problems with self-cohesion and emotion regulation. Psychotherapy can help the patient to link emotions, mental images narrative fragments, and behavior with a goal of increasing self-coherence.

Early attachment and caregiver experiences form the foundation for development of Identity and sense of self in relation to others. The infant/child requires recognition and marking of mental and emotional states in order to develop their own capacity to recognize, label, and ultimately modulate emotions [42,43,44]. This is foundational for the development of mentalization, the ability to appreciate and reflect on the mental states of others. The child who is not recognized and “kept in mind” is bound to have difficulties in self-recognition at multiple levels, and to be more vulnerable to states of disorganization, overwhelming affect, and experiences of aloneness. The individual develops the “inner working models” [45] or “implicit relational scripts” [46,47,48] from the relational experiences with early caregivers. These models define how one experiences oneself, and therefore how one acts, in relation to others. “These are the automatic procedural relationship rules that one lives by, generally without awareness: “This is how people experience me; this is how I am valued, or not valued, by others; this is how I tend to be treated; this is how my relationships go; this is what happens when I make my wishes and needs known; this is who I am” ([37], p. 29).

Implicit beliefs about self and relational patterns tend to be tenacious and resistant to change. Studies in social psychology have found that people tend to “self-verify,” seeking and selectively attending to feedback that confirms and reinforces the way that they already see themselves [49,50,51]. From an analytic perspective, Wachtel [52] described how people actively (although non-consciously) elicit confirmation of their self-concept by acting in ways that tend to evoke confirmatory responses; thus, the patient receives ongoing reinforcement that sustains his or her self-concept, even if that experience is terribly painful. The therapist engages the patient and disrupts the patient’s nonconscious relational scripts by behaving differently than was unconsciously expected, thus offering the possibility that the therapeutic relationship has the potential to facilitate change. The therapist also attends to the myriad of relational experiences that continue to reinforce and sustain the patient’s self-beliefs and behaviors. This means actively helping the patient to create new relational experiences that disrupt the repetitive elicitation of social feedback that confirms and sustains the patient’s maladaptive beliefs about self.

The therapist’s activity may include what are traditionally thought of as cognitive-behavioral strategies (e.g., problem-solving, modeling alternative ways of relating, exposure, role playing) to help the patient navigate difficult relational situations with the best possible chance to have positive and ‘script disrupting’ experiences. This is an iterative process, with repeated discrepant experiences with the therapist and with others gradually modifying the underlying dysfunctional relational scripts. It is especially critical in the psychotherapy of those who are vulnerable to suicidal states, for whom the repeated nonconscious evocation of confirmatory negative interpersonal experiences can quickly lead to turmoil and despair, making suicidal behavior more likely. This will be discussed further in the discussion below about the therapeutic alliance.

### 3.7. The Emergence and Recognition of the Patient’s Genuine Capacities

The emerging recognition and appreciation of one’s genuine capacities is a developmental process that can be derailed through trauma or other negative developmental experiences [37,53]. Loewald [54] described the importance of the analyst in this regard, holding and communicating a vision of the patient as who he or she is, as well as the “more” that he or she can come to be. Patients who are vulnerable to suicidal states often have anxiety associated with experiencing themselves positively, with harsh self-attack and automatic avoidant defenses that keep them from fully taking in positive relational and other experiences. The therapist, by noticing this nonconscious avoidance, calls attention to and thereby disrupts these defenses in a way that gives the patient an opportunity to bear the anxiety and conflict associated with experiencing oneself more fully. At times, this can mean integrating more formal exposure into the treatment (e.g., “I noticed when you talked about the ‘A’ that you got in class, you looked away and started to mumble…What were you feeling? Maybe you can say it again but try to look right at me and use a strong voice. Let’s see if you can sit with the feelings that this brings up.”). Over time, the anxiety begins to diminish, and the patient gradually begins to feel more comfortable with a more complete experience of himself or herself. This creates a positive feedback loop as the patient also begins to act in a way that is consistent with his or her emerging identity and thus is more likely to elicit, and attend to, confirmatory feedback (53), which has been described as a renegotiation of identity [50].

### 3.8. Continuity and Coherence

Continuity is the sense of being the same person with linked experiences over time; coherence has to do with linking experiences, roles and beliefs as part of an integrated whole [35]. Patients with Borderline Personality Disorder (BPD) have been described as being vulnerable to a sudden loss of continuity—the loss of evocative memory—that leaves them unable to evoke soothing introjects and acutely vulnerable to experiences of aloneness and suicidal desperation [21]. Maltsberger [27] envisioned the “descent into suicide” as an overwhelming, unbearable flooding of negative affect that leads to a breakup of self-cohesion, to which those who do not have a strong base of continuity of self are particularly vulnerable.

Severe distress becomes unbearable when one loses the capacity to experience connection with others, a state of excruciating aloneness. This becomes even more dire “with the loss of a continuous thread of being that mitigates the sense that now is all there is, that pain will be forever, that there is no hope of moving forward because there is no forward, and that suicide might be the only option” ([37], p. 34). Recurrent affective dysregulation, mood changes and psychotic experiences can themselves be experienced as traumatic and are disruptive to developing and sustaining continuity and coherence [19].

What this means is that the therapist should consider a sense of continuity and coherence a treatment goal rather than an expectation [37]. The therapist, at times, may need to painstakingly help the patient link experiences with each other, reminding the patient that aspects of the past continue to influence the present, and even helping the patient to link dissociated experiences day-to-day. Some patients are vulnerable to feeling suddenly disconnected, alone and desperate, without any sense that something may have happened to cause their change in mental state, and the timeless sense that they will never feel better again. At these moments, the therapist may help the patient through these experiences by inter-session contact of some kind (e.g., phone, email, voice mail), and also by helping the patient to connect feelings with experiences to begin to understand the transient vulnerability to these dissociated states.

### 3.9. Attention to the Therapeutic Alliance

Building and maintaining a strong therapeutic alliance is a cornerstone of psychotherapy with suicidal patients, but it is fraught with significant challenges (see [55,56]). Psychotherapy often involves strong transference–countertransference reactions to which both patient and therapist contribute. Patients often come to the encounter with intense affects, and negative relational expectations. Negative therapeutic reactions and treatment crises are common, and often inculcate a fear of catastrophic consequences. Dealing with these inherent difficulties is a challenge for both patient and therapist and is critical to the success of the therapy [57,58,59].

A related challenge for the therapist is the need to move flexibly between empathic listening and an ongoing suicide risk assessment [55,56]. The therapist shifts between creating a psychotherapeutic space and observing the patient objectively. When there is a change in the patient’s emotional state or degree of relatedness, the therapist begins an assessment of the patient’s safety, the strength of the alliance, and whether there is a need for other safety interventions. Anxiety about the patient’s safety can paradoxically take the therapist away from a stance of empathic attunement.

Suicidal patients generally have a harshly self-critical internal narrative, and an underlying expectation of negative interpersonal experiences that confirm their bad sense of self. In this context, the therapist needs to be actively affirmative and validating in order to achieve "functional neutrality" [60]. One issue on which psychodynamics differ is how to conceptualize and to manage the patient’s aggression and negative transference. Transference Focused Psychotherapy (TFP), for example, prioritizes interpreting aggressive, split off, unintegrated parts, and considers this an essential element even early in the course of treatment [61]. Our approach prioritizes attunement to the patient’s internal experience (as experienced by the patient), which we see as essential to strengthening the alliance and helping the patient with the experience of aloneness. We conceptualize aggression as most often a secondary response to the patient’s internal state of desperation. A metaphor that we find helpful is that of a person who is drowning, unable to breathe, and who desperately grasps for a life raft, pushing past and inadvertently injuring another person. One could take up the aggression as primary, but we do not feel that this is closest to the patient’s experience, for whom the lashing out came from a desperate effort to breathe and to stay alive. This is very much akin to the suicidal patient’s desperate experience of aloneness, abandonment, and other unbearable emotional experiences, which leads to an urgent need for relief/escape. We do not ignore aggression and take it up as needed to preserve the treatment and the frame. However, in our view, interpreting the negative transference, especially for early in treatment, can be experienced as distant and critical, which is antithetical to our goal of the early attunement and validation. In this way, we are very much aligned with the approach in Mentalization Based Therapy (MBT), with an emphasis on validation of the patient’s transference feeling, avoiding genetic interpretations that might be experienced as distancing and invalidating, exploring the transference, attending closely to the therapist’s contribution to the patient’s transference experience, and collaborative exploration [62].

The strength of the therapeutic alliance, including rupture and successful repair of the alliance, is a critical factor in the success of any psychotherapy (see [56,57,58]). In working with emotionally dysregulated patients, ruptures in the alliance can be particularly abrupt and severe—sometimes in response to something that occurs in the therapy, sometimes related to external stressors—with a risk of affective turmoil and potential for suicidal behavior. These ruptures can be extremely difficult for the therapist as well as for the patient, but if they can be repaired, they offer an opportunity for implicit relational learning and growth. In this context:
…the patient has the chance to discover experientially that an old relational pattern does not have to be repeated, and that the therapeutic relationship holds the promise of trying out, practicing and even solidifying something new. The patient also has the chance to learn that the therapist, and the relationship, can survive the intensity of the patient’s sudden withdrawal or affective storm. These moments are essential in gradually building trust, in strengthening the emotional bond between patient and therapist and in holding out the hope that the relationship might actually be of help.([56], p. 319)

In this reworking of the script, the patient has an opportunity to experience the therapist as caring, and himself or herself as being worthy of being cared about. Working through ruptures often requires that the patient experience something different from the therapist, in some way above and beyond what the patient expected. This might be in the form of a self-disclosure (e.g., the therapist might acknowledge an unempathetic reaction, and share that something about what the patient had said reminded him or her of a troubling experience), an action (e.g., having the patient stay a few minutes beyond the hour because of the importance of the patient’s association), or some other spontaneous and genuine way of relating to the patient.

Stern et al. [46] described the therapeutic importance of “moments of meeting,” in which “the therapist must use a specific aspect of his or her individuality that carries a personal signature. The two are meeting as persons relatively unhidden by their usual therapeutic roles, for that moment” (p. 913). Linehan [4] used the term “radical genuineness” to capture the same experience. Hoffman [63], writing about psychoanalysis, captured the implicit meaning in the therapist’s doing something that is experienced as different in response to the patient’s need:
When the patient senses that the analyst, in becoming more personally expressive and involved, is departing from an internalized convention of some kind, the patient has reason to feel recognized in a special way, the deviation, whatever its content and whatever the nature of the pressure from the patient, may reflect an emotional engagement on the analyst’s part that is responsive in a unique way to this particular patient…there is something about the deviation itself, regardless of content, that has therapeutic potential (pp. 187–193).

Similarly, the patient may require measures that go beyond the traditional frame of the psychotherapy session to maintain or regain an empathic connection [64]. Interventions such as between session contact, offering additional sessions, and intermittent telephone, email or text support can help the patient cope with affective intensity when in crisis. These actions may also have internal meaning for the patient, serving to enhance internalization of the therapist as a positive introject and strengthen the therapeutic alliance [56]. Thoughtful flexibility regarding the treatment frame can be experienced as validating by the patient (i.e., the therapist cares enough to be flexible and is willing to prioritize the patient’s perceived needs) and often helps to strengthen and solidify the therapeutic alliance. In cases of severe personality disorders such as antisocial and severely narcissistic personality disorders, it may at times be beneficial to set and adhere to a more rigid frame (such as that advocated in TFP) [61]. On the other hand, a strict requirement for adherence to a set frame may evoke feelings of being controlled and entrapped, and in some circumstances can hinder the development of the alliance. In our view, it is important that the patient experiences agency in the treatment and feels that the therapist is listening and taking seriously his or her concerns, with some degree of flexibility. This is best accomplished through a sensitive and proactive collaborative negotiation with the patient around frame issues, with attention to the patient’s wishes and perceived needs, therapeutic goals, and the therapist’s personal limits.

Patients who have experienced traumatic affective experiences (not just overt trauma and neglect, but also truly overwhelming and unbearable affective experiences and psychotic states) are vulnerable to an erosion of the capacity to fully engage with sustaining relationships [19]. In this context, it can be too hard for the patient to sustain hope, and too easy to begin to give up on the ties to loved ones that are life-sustaining. If the patient can experience the therapist’s genuine affective involvement—at both a cognitive and implicit relational level—this can be the first step to regaining some hope for change. The therapist’s effort at repair, experienced as genuine and emotionally resonant by the patient, “is inviting the patient to hang on, to turn towards the relationship instead of away from it and to use the therapist’s help to find ways to cope without resorting to suicide” ([56], p. 320).

### 3.10. Attention to Countertransference

Psychotherapy with suicidal patients is inherently extremely challenging. First, the therapist must accept that despite his or her best efforts there is no way to definitively control the outcome, and a patient may complete suicide. The lack of control about a potentially catastrophic outcome for which the therapist may feel blamed engenders anxiety, fear, and other emotions that are inescapable. In addition, repeatedly sitting with a patient who is in terrible emotional pain, without the means to provide relief, is a very difficult emotional task for the therapist.

Hopelessness can be particularly difficult to bear, and the therapist may feel worn down over time, beginning to accede to the patient’s often stated belief that nothing can possibly be of help [55]. In most cases, this is a “countertransference hopelessness” which can unconsciously be communicated to the patient in verbal interaction and also in non-verbal cues. This then invites feelings of aloneness and abandonment in the patient, increasing hopelessness. The therapist can use his or her own reflective capacity, internal sustaining resources, knowledge about psychotherapeutic potential for change, and clinical experience to balance these reactions to the patient and to maintain hope. Consultation with a colleague or supervisor or a colleague can be effective in helping the therapist to regain balance and re-engage with the patient.

Suicidal patients are suffering terribly, and therapists are highly motivated to help, but often lack any tools to alleviate the suffering (at least in the immediate term). This can activate multiple affects in the therapist and trigger unconscious defense mechanisms. The therapist may feel suddenly anxious, overwhelmed, frustrated, and hopeless. The therapist’s automatic defenses against such feelings may include avoidance of affect; exclusive focus on symptoms rather than feelings; efforts to control the patient’s behavior, or an opposite lack of engagement in the patient’s decision-making; and denial about the seriousness of the patient’s distress. In their classic paper, Maltsberger and Buie [65] discuss the “countertransference hate” that can arise in the therapist. They describe two components: “malice” which is a more overt experience leading to expression of irritation or anger towards the patient, and the even more concerning “aversion”, which represents unconscious emotional withdrawal. It is the latter, aversion, that is most scary in terms of increasing potential suicide risk, as patients tend to be exquisitely attuned to the therapist’s emotional state and level of engagement. Unconscious withdrawal by the therapist may be experienced with an increase in aloneness and a sense of abandonment, which can put the patient at risk for despair and suicidal behavior. Again, discussion with colleagues and formal consultation can be extremely helpful for the therapist to get out of the intensity of the dyad and to explore his or her feelings about the patient and the treatment.

## 4. Conclusions

In this paper, the Boston Suicide Study Group has presented a theoretical and clinical approach to psychotherapy with suicidal patients. It is fundamentally a psychodynamic treatment, with an emphasis on the therapeutic alliance, unconscious and implicit relational processes, and the power of the therapeutic relationship to facilitate change in a long-term exploratory treatment. It is also integrative, incorporating ideas from DBT, MBT, and CBT, developmental and social psychology in a psychodynamic frame. There are ten key aspects of this integrative psychodynamic treatment: 1. approach to the patient in crisis; 2, instilling hope; 3. a focus on the patient’s internal affective experience; 4. attention to conscious and unconscious beliefs and fantasies; 5. improving affect tolerance; 6. development of narrative identity and modification of "relational scripts"; 7. facilitation of the emergence of the patient’s genuine capacities; 8. improving a sense of continuity and coherence; 9 attention to the therapeutic alliance; 10. attention to countertransference. These elements of treatment are overlapping and are not meant to be sequential, but they provide a “road map” of the various ways in which this psychotherapy facilitates a patient’s move from despair and suicidality to a more satisfying relational experience with others.

Engagement with an empathic therapist provides a secure environment for exploration of the suicidal patient’s internal subjective experience. The therapist helps to instill hope by bearing the patient’s pain while helping the patient to discover and mobilize a sense of agency, mourn what truly is not possible, re-engage in relationships, and begin to envision a life that is worth living. The therapist notices and disrupts the patient’s non-conscious defenses, leading to exposure to previously warded off feelings and greater affect tolerance. In listening to and engaging in the patient’s narrative, the therapist helps to co-construct a new narrative that is more empathic and less self-attacking, renegotiating narrative identity in a more realistic and positive way. The therapist facilitates the gradual emergence of the patient’s genuine capacities, modulating self-attack and disrupting avoidant defenses, and providing the patient an opportunity to bear the anxiety and conflict associated with experiencing oneself more fully. Great attention is paid to the therapeutic alliance and the therapist’s countertransference. 

The therapist engages with and disrupts the patient’s nonconscious relational scripts by behaving differently than unconsciously expected, thus offering the possibility that the therapeutic relationship has the potential to facilitate change. At the same time, the therapist actively supports the patient in creating new relational experiences that begin to elicit feedback that is discrepant with maladaptive core beliefs about self. Ruptures in the therapeutic alliance can be extremely painful, but repair of the alliance offers a further opportunity for implicit relational learning and growth.

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
