# Peer review of "Psychotherapy with Suicidal Patients: The Integrative Psychodynamic Approach of the Boston Suicide Study Group"

_1010-660X, 2019, doi:10.3390/medicina55060303_

Round 1

Reviewer 1 Report

This is an interesting review article on a highly relevant clinical subject. The authors – all highly qualified and distinguished in the field – provide a valuable overview on psychodynamic psychotherapy with suicidal patients primarily drawing from their extensive clinical experience as well as from relevant literature. The best parts of the article are some of the detailed discussions of clinically relevant issues in working with suicidal patients, such as countertransference problems or the focus on unbearable internal affective states of suicidal patients.

However, there are a couple of points that should be addressed in order to strengthen the manuscript:

·       The introduction reads more like a summary of the subsequent article than a proper introduction. There is no general framework to contextualize the conducted work and no reference to similar previous studies in this area.

·       Furthermore, a more concise delineation of the scope of the paper would be desirable. The authors suggest that they want to synthesize what is known about the psychotherapy of suicidal patients, particularly psychodynamic psychotherapies (“materials and methods” section). They apparently do not refer to empirical studies, where data is definitely scarce. But the clinically oriented literature in this field is quite rich and the authors are highly selective in the works they cite. This is not a problem per se in a predominantly clinical paper like this but should be stated clearer.

·       The notion of “psychodynamic psychotherapy” is used rather vaguely in the article. There are many important works by psychoanalytic therapists on work with suicidal patients (e.g. Kernberg, Gabbard, Bell,…) that are not considered by the authors. On the other hand they extensively reference DBT or other psychological therapies & theories (narrative identity,…). A more detailed definition of the scope of this notion would be helpful.

·       In this line, there are several references to the “integration” of non-psychodynamic techniques (affective regulation skills, exposure,…). Again, the question of what is specifically “psychodynamic” in this approach becomes uncertain.

·       The division between “results” and “discussion” seems artificial and not reader-friendly. The points of the “results” section would maybe best be turned into a table with the explication and discussion of the points in the running text.

·       The phrase: “People are complex, and the unconscious is not a rational place,…” (p.6, 243-244) seems rather colloquial.

·       The therapeutic stance to the treatment of suicidality in the context of severe personality disorders (e.g. BPD) advocated by the authors (e.g. in the section “continuity and coherence” or “attention to the therapeutic alliance”) is rather different from other psychodynamic approaches such as the one derived from the work of Otto Kernberg in Transference Focused Psychotherapy with regards to keeping the frame (e.g. inter-session contacts, self-disclosure). This would merit further discussion, especially since TFP (alongside MBT) is one of the very few psychodynamic therapies with empirical data to support its effectiveness.

Author Response

Thank you for your feedback about our manuscript.  We think that the changes that we have made in response to both reviewers has helped us to clarify a number of things and improve the pap

The abstract and introduction have been re-written to better frame what the article is about.  There is also greater clarity about the scope of the article, which is not to be a comprehensive review but rather a description of the Boston Suicide Study Group’s integrative psychodynamic approach.  The paper cites literature that supports and helps to explicate the thinking of the authors in conceptualizing their approach.

The revised manuscript clarifies that the treatment approach is fundamentally psychodynamic in nature, with an emphasis on the therapeutic alliance, unconscious and implicit relational processes, and the power of the therapeutic relationship to facilitate change in a long-term exploratory treatment. However, it is also integrative and draws extensively from DBT, CBT, developmental psychology and social psychology. 

With the permission of the Editor-in-Chief, the manuscript no longer has a "results" section. That format does not fit the nature of this contribution and we recognize that our original effort to fit the manuscript to that format was awkward and confusing.

The phrase "people are complex, and the unconscious is not a rational place,…” (p.6, 243-244) has been changed to: “These fantasies are often not rational and in contradiction with the patient’s known reality.” (p. 5, 211-212)

The manuscript now clarifies that the approach of the Boston Suicide Study Group is quite different from that of TFP.  We emphasize validation and attention to the alliance and discourage early transference interpretations about the negative transference.  In this way our approach is very much aligned with that of MBT. (See lines 396-418 and 454-470 for these changes)

Reviewer 2 Report

Dear authors

your paper deals with a very interesting issue and extensively describes persepctives and implications of a psychodinamic approach with suicidal patient. I would suggest the following revisions so as to improve the paper:

1) Title and type of study: My major concern is about the definition of the study as a review. The reader could misunderstood, and I, as reviewer, I disagree with such a definition. Your paper is neither a systematic review, nor a narrative review. It may be consider a theoretical paper "based on the authors’ extensive clinical work with suicidal patients" (line 73). I would suggest how to present the paper and define the type of the study.

2) Methodology: Accordingly to the previous revision, the methodology has to be reviewed and reconsidered. I would  suggest to implement the description of "Materials and Methods" section, and so the rationale of presenting ("Results" section) your reflection on the topic.

3) Background: By assuming that psychodinamic psychotherapy is a wiide range of perspective, the "Introduction" section does not really define your own perspective and, especially the theoretical and clinical sources that may be consider the foundation of your practice and so of your reflections.

4) Theorethical Reasoning: I would kindly ask you to generically revise your reasoning and especially statements by increasing and specifying the references you are based them upon (accordingly to previous revision).

I hope these suggestions may support you in improving your paper.

Regards

Author Response

Thank you for your feedback about our manuscript.  We think that the feedback from both reviewers has helped us to clarify and improve the paper. Some of our response to your feedback has been incorporated into our response to review her #1, who made similar requests for areas of revision.  Regarding your specific comments:

1.  Title and type of study: The authors agree that this paper is not a comprehensive review of the literature.  Rather it is a theoretical paper that synthesizes the authors’ thinking about and approach to the critical aspects of psychotherapy with suicidal patients.  The approach is psychodynamic in nature, with an emphasis on the therapeutic alliance, unconscious and implicit relational processes, and an emphasis on change that can occur with a long-term exploratory treatment relationship.  However, it is also integrative, drawing extensively on DBT, MBT, CBT, developmental and social psychology.

2.    With permission from the Editor-in-Chief, the resubmitted manuscript does not have a methodology section.

3.   Background: See response #1.

4.  Theoretical reasoning: The authors have clarified their arguments wherever possible. Specifically, the revised manuscript clarifies that the treatment approach is fundamentally psychodynamic in nature, while also integrating ideas and techniques from other schools of thought. Respectfully, the authors do not agree with a need to "generically revise" their reasoning across the board, but we do believe that the changes in the re-submitted manuscript respond to the spirit of the constructive feedback from both reviewers.

Round 2

Reviewer 1 Report

The authors convincingly addressed all points raised. I have no further comments. This is a very valuable contribution to psychodynamic approaches to the treatment of suicidal patients.

Reviewer 2 Report

Dear authors

the paper has been improved. And it may be considered for publication.